# CellMixS: quantifying and visualizing batch effects in single-cell RNA-seq data

Almut Lütge[1,2] ⓘ, Joanna Zyprych-Walczak[3], Urszula Brykczynska Kunzmann[4], Helena L Crowell[1,2], Daniela Calini[5], Dheeraj Malhotra[5] ⓘ, Charlotte Soneson[2,4] ⓘ, Mark D Robinson[1,2] ⓘ

A key challenge in single-cell RNA-sequencing (scRNA-seq) data analysis is batch effects that can obscure the biological signal of interest. Although there are various tools and methods to correct for batch effects, their performance can vary. Therefore, it is important to understand how batch effects manifest to adjust for them. Here, we systematically explore batch effects across various scRNA-seq datasets according to magnitude, cell type specificity, and complexity. We developed a cell-specific mixing score (cms) that quantifies mixing of cells from multiple batches. By considering distance distributions, the score is able to detect local batch bias as well as differentiate between unbalanced batches and systematic differences between cells of the same cell type. We compare metrics in scRNA-seq data using real and synthetic datasets and whereas these metrics target the same question and are used interchangeably, we find differences in scalability, sensitivity, and ability to handle differentially abundant cell types. We find that cell-specific metrics outperform cell type–specific and global metrics and recommend them for both method benchmarks and batch exploration.

## Introduction

Batch effects and data integration are well-known challenges in single-cell RNA-sequencing (scRNA-seq) data analysis and a variety of tools have been developed to overcome them (1, 2, 3, 4 *Preprint*, 5). Often, the terms data integration and batch effect removal are used interchangeably, but recently they have been distinguished by complexity: batch correction refers to the removal of simple biases (e.g., between datasets from the same laboratory) and data integration refers to matching data with nested layers of unwanted variation (e.g., data from multiple laboratories or protocols) (4 *Preprint*). Notably, benchmark studies of batch correction and data integration methods showed differences in performance on datasets depending on the batch effect complexity (4 *Preprint*, 5, 6 *Preprint*).

Although this underscores the need to know the nature and source of a batch effect, there is no systematic understanding of how batch effects manifest in single cell data and how much they can vary. Key aspects that characterize batch complexity are the strength of the batch effect with respect to other sources of variation and the cell type specificity, for example, storage conditions could cause higher stress on particular cell types or some cell types could generally exhibit more variation in their expression profiles.

Synthetic data that provide a ground truth are important for method evaluation (7). So far, simulations do not appear to reflect the batch complexities of real data because integration results based on simulations differed substantially from those based on real datasets (2, 4 *Preprint*, 5). In particular, recent studies (4 *Preprint*, 5) introduced batch effects by multiplying the mean counts of a certain batch by gene-wise batch factors sampled from a log-normal distribution. We highlight below that batch effects generally come with gene- and cell type–specific log fold changes, and we thus, generated our own synthetic datasets that reflected cell type specificity and covariance of real data batches.

Data integration aims to ensure consistent clustering across different batches. Given the frequent use and varying performance of integration methods (4 *Preprint*, 5, 6 *Preprint*), it is also important to understand the impact of data integration on a per-dataset level. There are different ways to assess the presence and strength of a batch effect before and after integration. A useful qualitative way is by visualization, by representing a dataset's main axes of variability in a two-dimensional space (e.g., using tSNE (8) UMAP (9)). However, in datasets with multiple sources of variation and in comparative settings, quantitative metrics are necessary to summarize the batch effect. A variety of metrics have been proposed to quantify batch effects or more specifically, the mixing of cells from multiples batches (see Table 1). Some assign a score to each cell based on neighbourhood mixing (cell specific), whereas other metrics assess the batch mixing within each cell type (cell type specific) or summarize a general batch mixing for the entire dataset (global); indeed, cell-specific scores can be aggregated to the subpopulation

[1]Department of Molecular Life Sciences, University of Zurich, Zurich, Switzerland   [2]SIB Swiss Institute of Bioinformatics, University of Zurich, Zurich, Switzerland   [3]Department of Mathematical and Statistical Methods, Poznan University of Life Sciences, Poznań, Poland   [4]Friedrich Miescher Institute for Biomedical Research, Basel, Switzerland   [5]F. Hoffmann-LaRoche Ltd, Pharma Research and Early Development, Neuroscience, Ophthalmologyand Rare Diseases, Roche Innovation Center Basel, Basel, Switzerland

Correspondence: mark.robinson@mls.uzh.ch

Table 1.  Batch mixing metrics: short summaries of metrics included in the benchmark.

| Metric | Level | Basis | Short description | Interpretation |
|---|---|---|---|---|
| Cell-specific Mixing Score (cms) | Cell | knn, pca | Test for whether distance distributions from a neighbourhood are batch specific | *P*-value: Probability to observe as large differences in distance distributions assuming the same underlying distribution |
| Local Inverse Simpson Index (lisi) | Cell | knn | Inverse of the sum of batch probabilities within weighted knn | Effective number of batches in neighbourhood |
| Entropy | Cell | knn | Sum of the products of the batch probabilities and their log within each cell's knn | Randomness in the data according to the batch variable |
| Mixing metric (mm) | Cell | knn | Median position of the fifth cell from each batch within its knn | Number of cells within knn until each batch is represented by five cells |
| Graph connectivity (graph) | Cell type | knn-graph | Fraction of directly connected cells within cell type graphs | Proportion of non-distorted cell type relationships |
| k-nearest neighbour Batch effect test (kBet) | Cell type | knn | Test for equal batch proportions within a random cell's knn | *P*-value: Probability to observe as large differences in batch proportions assuming the same underlying proportions |
| Average silhouette width (asw) | Cell type | pca | Average relationship of within and between batch-cluster distances for each cell type | Indication of how well clusters are separated |
| Principal component regression (pcr) | Global | pca | Correlation of the batch variable with principal components weighted by their variance attributes | Proportion of variance attributed to batch |

level and all scores can be reported at a global level. A common strategy of cell-specific metrics is to estimate the batch effect within each cell's k-nearest neighbourhood (knn), for example, based on entropy. Most cell type–specific metrics use the consistency or distortion of cell type clusters as a measure for the batch effect, for example, based on the silhouette width. The only global metric proposed so far is based on correlation of the batch effect with principal components and their corresponding variance. Despite the fact that these metrics serve the same purpose of quantifying the magnitude of the batch effect, they showed distinct differences in their rankings and results when applied to the same datasets (4 *Preprint*, 5). Thus, it is important to understand differences between the metrics and whether certain metrics are advantageous in specific contexts.

Here, we developed the cell-specific mixing score (cms) to detect bias according to a batch variable within scRNA-seq data. cms compares batch-specific distance distributions of each cell's knn. We characterized nine batch effects with different sources of known bias to explore the range of batch effects in real scRNA-seq data. Next, we used these batch characteristics of real datasets to extend and adjust the muscat simulation framework (10) to integrate cell type–specific batch effects. We used these synthetic datasets to benchmark metrics along several dimensions: scaling with the batch strength, scaling with randomness in the batch variable, sensitivity to detect batch effects and robustness to unbalanced batches.

## Results

### Characterization of batch effects

We explored batch effects across various sources and magnitudes to be able to generate realistic synthetic data. In total, we analysed

seven datasets (some with multiple sources of batch effects) with batch effects related to differences between patients, media storage, and use of sequencing protocols (see Fig S1 and Table S1). Initially, we focused on quantifying batch strength relative to other sources of variation, such as cell type specificity.

For each gene (in each dataset), we fit a linear mixed model to partition the variance (11) attributable to batch, cell type and an interaction of batch and cell type:

$$Y_g = \mu + X_p \alpha_{pg} + X_b \beta_{bg} + X_{p:b} \gamma_{(p:b)g} + \varepsilon_g, \tag{1}$$

where $Y_g$ is the (normalized and log-transformed) expression of gene $g$ across all cells (for a dataset), $\mu$ is the baseline expression, $X_p$, $X_b$, and $X_{p:b}$ are design matrices for the (random) cell types, batches and interactions, $\alpha_{pg} \sim N(0, \sigma^2_{pg})$, $\beta_{bg} \sim N(0, \sigma^2_{bg})$, and $\gamma_{pg} \sim N(0, \sigma^2_{(p:b)g})$ represent the corresponding random effects and, $\varepsilon_i \sim N(0, \sigma^2_g)$ represents the remaining error.

As shown in Fig 1A, batch effects attributed to sequencing protocols (cellbench, hca, pancreas) showed the highest average per cent variance explained by the batch effect (PVE Batch), according to their highly variable genes (HVGs). Batch effects attributed to sequencing protocols also showed the highest number of genes with a high PVE-Batch. In contrast, in datasets with batch effects attributed to media storage (csf_media, pbmc_roche, pbmc2_media) or patients (csf_patients, pbmc2_pat, kang), most genes showed a high percentage of variance explained by the cell type effect (PVE-Celltype), whereas the batch effect influenced a smaller subset of the genes. This is in line with our expectations: storage conditions and differences between patients affect specific genes, whereas sequencing protocols have a broader effect. In kang and pbmc_roche, only a few genes showed a high PVE-Batch. Both datasets also showed a mild batch effect, based on visual inspection of the tSNE (see Fig S2). We also find clear differences in the per cent variance explained by the interaction effect (PVE-Int) of

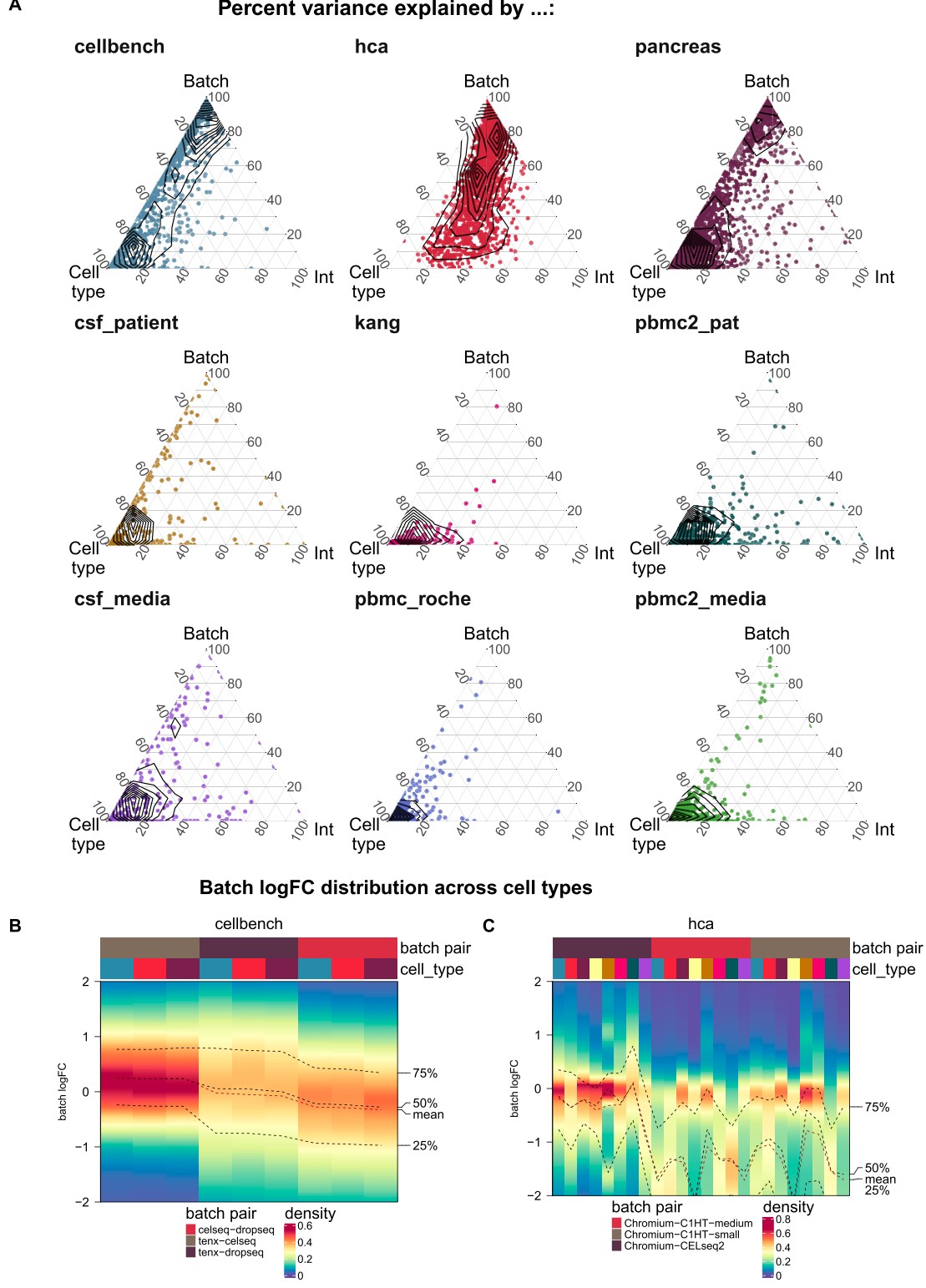

**Figure 1. Batch characterization.**
**(A)** Gene-wise variance partitioning across datasets. Each dot in each ternary plot represents a gene's relative amount of variance explained (by batch, cell type or interaction). **(B, C)** Batch logFC distribution by cell type and batch effect in the cellbench and hca datasets, respectively. Each column represents a density plot of the estimated logFCs for a batch/cell type combination. Dotted lines indicate the mean, 25%, 50% and 75% percentiles.

the cell type and batch effect (int). For some datasets, such as pbmc2_pat, there are more genes with a high PVE-Int than PVE-Batch, whereas for other datasets, for example, pbmc2_media, most batch-associated genes have the largest part of their variance explained by batch. In the cellbench dataset, only a minority of HVGs had some PVE-Int, whereas in the hca dataset, almost all HVG genes showed some percentage of variance attributed to the interaction. This aligns with findings from batch-associated log-fold change (logFC) distributions. In the cellbench dataset, the logFC distributions differ mostly between, but not within batches (see Fig 1B), indicating little to no cell type specificity of the batch effect. In the hca dataset, the logFC distributions also differ between cell types of the same batch (see Fig 1C), indicating high cell type specificity.

### Cell-specific mixing score (cms)

We developed a cell-specific mixing score to evaluate local batch mixing, independently of clustering or cell type assignment. For each cell, batch-wise distance distributions to the cell's knn are retrieved. cms scores the null hypothesis that the distances originate from the same distribution (across batches) using the Anderson–Darling test (12) (see Fig 2A–D). To avoid the curse of dimensionality, distances can be derived from Euclidean distances in principal component analysis (PCA) space or any other low dimensional representation with meaningful distances. Because the Anderson–Darling test is based on distances, and not directly on the number of cells per batch, it is robust to differentially abundant batches. The score for each cell can be interpreted as a *P*-value, that is, the probability of observing such deviations in the batch-specific distance distributions by chance (assuming they are all derived from the same distribution). Thus, enrichment of low *P*-values is indicative of poor batch mixing and a dataset with randomly shuffled batch labels should yield uniformly distributed *P*-values (see Fig 2E). A key parameter for the cms calculation is the neighbourhood size, $k$, which determines which cells to include in distance computations. We considered three ways of defining this neighbourhood: cms_default uses the same $k$ for all cells, with a larger $k$ oriented towards global structures and smaller $k$ to detect local effects; cms_kmin uses a dynamic density-based neighbourhood, which takes into account that the optimal size of the neighbourhood can vary within a dataset; cms_bmin uses a defined minimum number of cells per batch and includes cells from other batches until it reaches this minimum for all batches.

### Comparison of batch mixing metrics (mms)

A variety of metrics have been proposed to quantify batch effects. Here, we systematically compare eight metrics, including the cell-specific mixing score (cms), local inverse Simpson index (lisi) (3), Shannon's entropy (entropy) (6 *Preprint*), mm (13), graph connectivity (graph) (4 *Preprint*), knn batch effect test (kBet) (2), average silhouette width (asw) (4 *Preprint*, 5) and principal component regression (pcr) (4 *Preprint*). Short summaries of these metrics are shown in Table 1 and detailed descriptions can be found in the Materials and Methods section. In addition we tested different variants for two of these metrics (cms and lisi). For cms, we tested

three ways of defining the neighbourhood (cms_default, cms_kmin, and cms_bmin, as discussed above). For the inverse Simpson index, we tested different ways to weight the neighbourhood: lisi uses Gaussian kernel-based distributions for distance-based neighbourhood weightening, wisi uses Euclidean distances to weight neighbourhoods, and isi does not weight neighbourhoods at all.

Altogether, we designed five benchmark tasks to cover the most relevant use cases of these metrics (see Table 2 for short descriptions). One major application of these metrics is to assess the severity of a batch effect and thus reflect the level of confounding. For example, a larger score should result from a stronger batch effect across datasets (Task 1). Metrics should be able to distinguish between a batch effect and random mixing of batches (Task 2), while also having high sensitivity to detect systematic differences related to the batch variable (Task 3). A further application is to compare batch removal or data integration methods, before and after; metrics that are well suited for this task should scale with the strength of the batch effect within the same dataset (Task 3) and should be stable towards changes in composition between batches (Task 4). Computational time and memory footprint can also be important considerations (Task 5).

### Task 1: reflection of batch characteristics

In this task, we tested a metric's ability to discriminate between a strong and a mild batch effect across datasets. This is an important feature of these metrics as the impact of a batch effect is context-specific and depends on how strongly interesting data characteristics are confounded. To test this, we used the batch characteristics and datasets explored above. In particular, we used the average PVE-Batch across all genes and the proportion of DE genes between batches as a surrogate for the strength of the batch effect across datasets (see Fig 3). We then calculated the Spearman correlation between these batch strength measures and the aggregated metric scores. Scores were aggregated using the mean score across all cells, thus to aggregate cell type–specific metrics, we used the mean (weighted by the number of cells). Cell-specific metrics, with the exception of isi, showed high correlation with at least one of the surrogates. Most cell-specific metrics showed a plateau in their score towards higher batch strength, suggesting a maximal score has been reached and thus they cannot further discern the strength of a batch effect. In Fig S2, we show 2D tSNE projections of all datasets ordered by their percentage of DE genes between batches. All datasets except the kang and pbmc_roche dataset exhibit clear batch effects that can easily be identified by visualization, where most neighbourhoods consist of cells from the same batch. Although cell-specific metrics that only consider each cell's neighbourhood get saturated at their nominal minimum in these cases (from the csf_patient dataset onwards in Fig S2), their summarized score still reflects the overall order of datasets based on batch strength measures. asw and pcr show lower overall correlation with batch magnitude, but are nonetheless able to distinguish mild batch effects (almost no DE genes between batches; low average PVE-Batch) from strong batch effects. In a relative sense, kBet and graph did not correlate well with batch strength, that is, their absolute score does not appear to reflect batch strength across datasets as defined here.

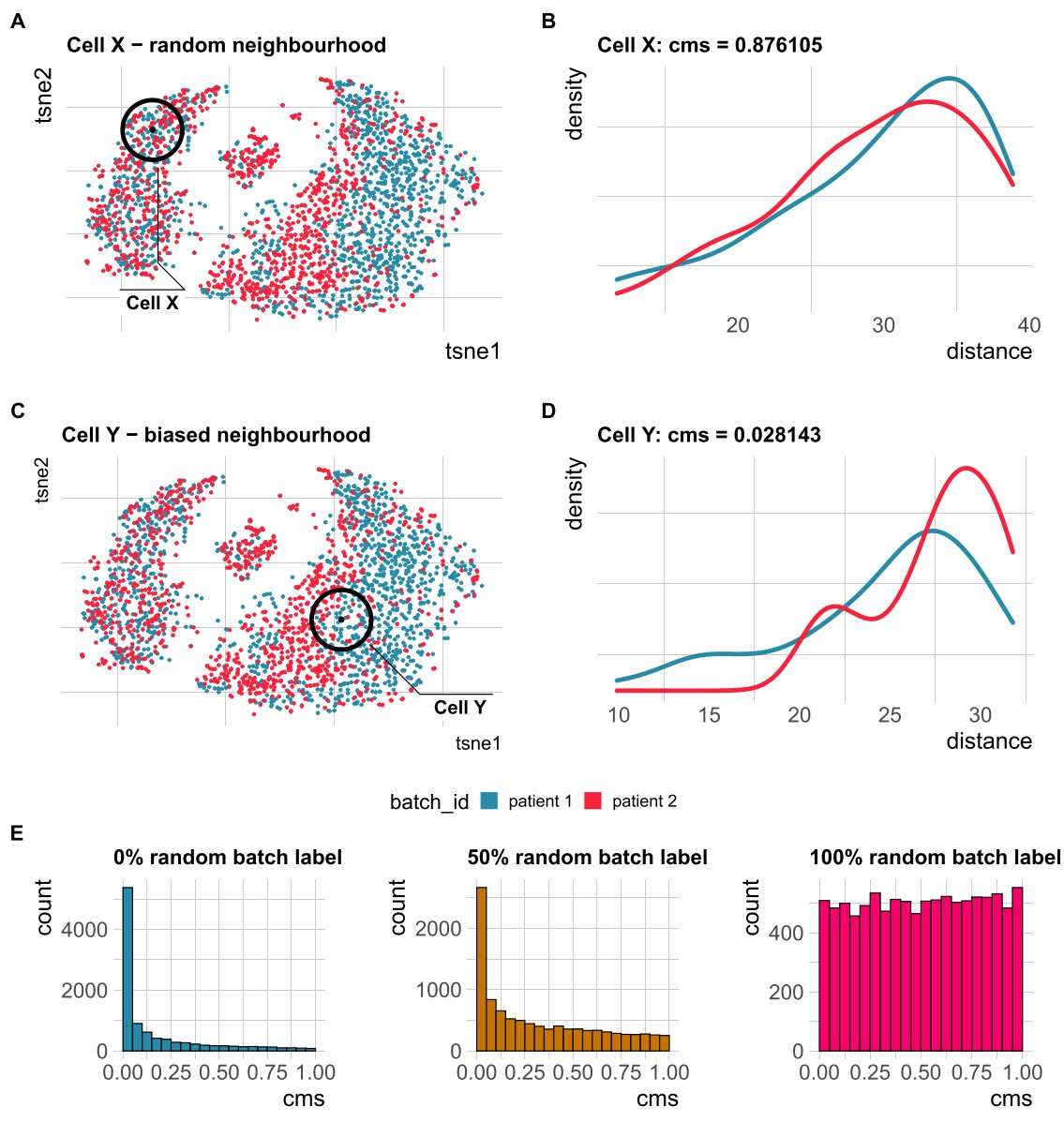

**Figure 2. Cell-specific mixing score cms.**
**(A)** A two-dimensional (2D) tSNE projection of synthetic single-cell RNA-sequencing data with two batches and a cell type–specific batch effect. Cell X is surrounded by an equally mixed neighbourhood with no batch effect. **(B)** Batch-specific Euclidean distance distributions for all cells within $k$ nearest neighbours (knn) of cell X in principal component analysis space (PC1-PC10). cms (Anderson–Darling $P$-value) = 0.88. **(C)** A 2D tSNE projection of synthetic single-cell RNA-sequencing data with two batches and a cell type–specific batch effect. Cell Y is surrounded by a biased neighbourhood with a batch effect. Distances towards cells from the "red" batch are larger than those towards cells from the "blue" batch. **(D)** Batch-specific Euclidean distance distributions for all cells within knn of cell Y in principal component analysis space (PC1-PC10). cms = 0.03. **(E)** Score distribution of cms in a dataset with four batches with 0%, 50% and 100% of batch labels permuted (see benchmark Task 2).

## Task 2: scaling with batch label permutation

All metrics have been proposed to detect transcriptional signatures associated with the batch variable in scRNA-seq data. Therefore, they should decrease/increase (depending on the directionality of the metric) as batch labels are randomized, and achieve a minimum/maximum when all labels are assigned at random; that is, random labels serve as a negative control. We used two real datasets, one with two clearly separated batches and one with a moderate batch effect, and permuted a percentage of the batch labels (from 0 to 100% in steps of 10%, see Fig 4A). For each metric,

we calculated the Spearman correlation of the mean score with the percentage of permuted labels. All metrics, except asw and graph, showed a high correlation in both datasets (see Figs 4B and S3). As expected, scores for the fully randomized labels reveal a flat $P$-value distribution for the cms scores, the effective number of batches as average lisi scores, an average entropy close to 1 and a pcr score of 0 (see Fig S4A–F). The asw score shows high correlation with random label percentage in the dataset with a strong batch effect (see Fig 4B), but low correlation (R = 0.16) in the dataset with a moderate batch effect (see Fig S3). For this dataset, the asw score is almost unchanged with increased randomness; we presume it fails

**Table 2. Summary benchmark tasks: In total, five tasks were designed to evaluate scaling, sensitivity, stability, correspondence to real data batch characteristics, and computational time and memory.**

| Name | Measure | Aim |
|---|---|---|
| Task 1: Batch characteristics | Spearman correlation of metrics with surrogates of batch strength (e.g., PVE-Batch and proportion of DE genes between batches) across datasets | Test whether metrics reflect batch strength/ confounding across datasets |
| Task 2: Scaling with batch label permutation | Spearman correlation of metrics with the percentage of randomly permuted batch label | Serves as a negative control and determines whether metrics scale with randomness |
| Task 3: Scaling with batch strength and detection limits | Spearman correlation of metrics with the batch logFC in simulation series on the same dataset; minimal batch logFC that is recognized from the metrics as batch effect | Test whether metrics scale with (synthetic) batch strength; Estimate lower limit of batch detection |
| Task 4: Unbalanced batches | Reaction of metrics to imbalance cell type abundance within the same dataset | Test sensitivity towards imbalance of cell type abundance |
| Task 5: Computational time and memory | CPU time and memory usage according to number of cells and number of genes | Assess computational cost of metrics |

For each task, different datasets (synthetic, semi-synthetic, or real) were used.

more because of its limited sensitivity than its inability to scale with randomness. In contrast, graph fails in both scenarios (see Figs 4B and S3) because it is a metric to test the connectivity of cell type clusters and the cluster membership is unchanged by permuting batch labels; in fact, graph is the only metric that is fully independent of the batch label.

## Task 3: scaling with batch strength and detection limits in synthetic data

To test overall sensitivity to detect batch effects, we used a simulation series with increasing batch effects. Within each series, we sampled gene counts $Y_{gcb}$ from a negative binomial distribution with mean, $\mu_{gcb}$, and dispersion, $\phi_g$:

$$Y_{gcb} \sim NB(\mu_{gcb}, \phi_g), \qquad (2)$$

for gene $g$ in cell $c$ from batch $b$. Similar to Equation (1), the means are adjusted for each combination of batch and cell type (cluster), giving an intercept parameter, $\beta_{0g}^t$, which represents the baseline relative expression level for cell type $t$. For each series, a reference dataset is used to estimate $\phi_g$, $\beta_{0g}^t$, the (effective) library size $\lambda_c$ and logFC parameters $\beta_g^{tb}$ for each batch. To modulate the batch strength, these logFCs are multiplied by a factor $\theta_b$, such that:

$$\mu_{gcb} = exp\left(\beta_{0g}^t\right) * \lambda_c * 2^{\beta_g^{tb} \cdot \theta_b}. \qquad (3)$$

Here, we multiplied batch-associated logFCs by a series of 13 factors $\theta_b \in [0, 4]$ resulting in no batch effect, and both an attenuated and increased batch effect relative to that estimated from the reference dataset (see Fig 5A).

In total, we generated simulation series across seven reference datasets (13 different multiplicative logFC factors for each dataset). We computed Spearman correlation coefficients of the metrics against the multiplicative batch factor $\theta_b$. In general, cell-specific and global metrics showed high concordance with the true relative batch strength of the simulations, revealing correlation coefficients ≥0.9 (see Figs 5B and S5). Cell type–specific metrics (kBet, asw, graph) showed

a dataset-dependent performance with correlations ≤0.7 for several datasets. On average, kBet showed the lowest correlation with the true (relative) strength of the batch effect. In some simulation series, scores of neighbourhood-based metrics (cms, lisi, and kBet) were saturated near their maximum before the highest batch logFC was reached. Whereas cms and lisi flattened out and still showed high overall correlation in these cases, kBet was already saturated at low batch logFCs, which was reflected in a lower correlation coefficient (see Fig S5). To compare a metric's ability to detect subtle batch effects (detection sensitivity), we evaluated at which relative batch strength (multiplicative factor) the scores started to differ recognizably from their value in the batch-free control dataset. We defined this batch detection limit as the lowest multiplicative factor that led to a metric score with more than 10% of the score's overall range from the batch-free score. Because batch logFCs in these simulation series were relative towards the logFCs of the reference datasets, these limits cannot be interpreted as absolute limits, but only in a comparative way across metrics. cms, kBet, lisi, and entropy showed the lowest batch detection limits and therefore highest sensitivity, whereas pcr, graph, and asw showed the lowest sensitivity (see Figs 5C and S5). In general, all cell-specific metrics except mm showed strong sensitivity and also scaled with batch strength. For pcr and kBet, there is a clear trade-off between sensitivity and scaling (see Fig 5D).

## Task 4: unbalanced batches with differential cell type abundance

Another important aspect of quantifying batch effects is to ensure that metrics are not sensitive to differential abundance of cell types, since these can often be present even in the absence of batch effects. To test this, we randomly removed an increasing number of cells from one batch in one cell type in three simulated datasets: the first dataset is effectively "batch-free," the second has moderate batch effect with cells still clustering by cell type and the third one has clearly separated batches. A performant metric should not change by removal of cells because the batch effect itself remains unchanged. Fig 6A shows how metric scores changed in the dataset with a moderate batch effect starting with approximately equal batch proportions (0% unbalanced) and finishing with one cell type uniquely present in one batch (100%

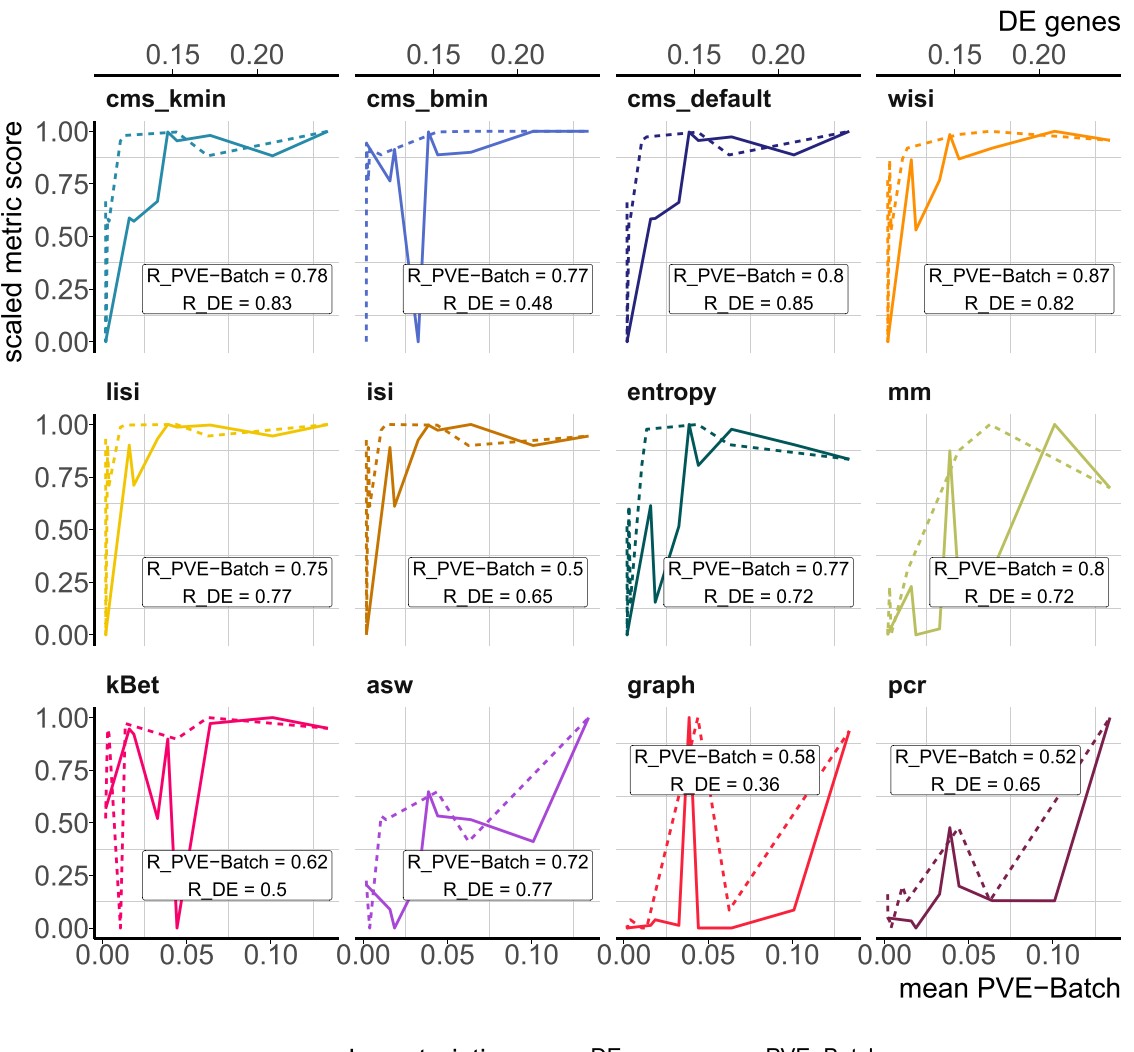

**Figure 3. Task 1—Reflection of batch characteristics.**
Metric scores versus (surrogate) batch strength across the real datasets. Summarized metric scores (y-axis) are compared with the proportion of DE genes (top x-axis, solid line) and the mean PVE-Batch (bottom x-axis, dashed line) per dataset. Datasets with a stronger batch effect (high percentage of DE genes/mean PVE-Batch) are expected to show a higher overall metric score than datasets with mild batch effects (low percentage of DE genes/mean PVE-Batch). Spearman correlation coefficients of metrics against the two batch strength measures are shown (R_PVE-Batch, R_DE) in the text boxes for each metric and evaluated in Task 1. Metric scores were standardized by subtraction of their minimum and division of their range (maximum−minimum) across datasets. Directions were adjusted when necessary, such that all scores increase with batch strength.

unbalanced). We defined the proportion of removed cells that caused a divergence in the metric score by more than 5% of its score in a balanced setting (equal number of cells per batch in all cell types) as the imbalance limit. An imbalance limit of 1 corresponds to a metric that showed a stable score until all cells from one batch within one cell type were removed. Fig 6B shows the imbalance limits of all metrics across datasets. pcr remained unchanged for all three batch scenarios. cms scores only exhibited minor changes if one batch is completely removed and remained otherwise stable. asw and mm only showed minor divergence from their original score. lisi scores and entropy showed a clear difference already at 30–50% (cells from one batch removed) for the

moderate batch and also diverged with 70% removed in the batch-free dataset. kBet was stable in the batch-free setting, but showed low imbalance limits when the batch effect was moderate or strong. graph remained unchanged for all datasets, except in the dataset with separated batches.

**Task 5: computational time and memory consumption**

We evaluated CPU time (User time + system time) and memory usage (maximum resident set size: RSS) in two synthetic datasets with different numbers of cells (68,472 and 80,768) and genes (8,331 and 23,381). We downsampled datasets in steps of 20% to explore

**A**

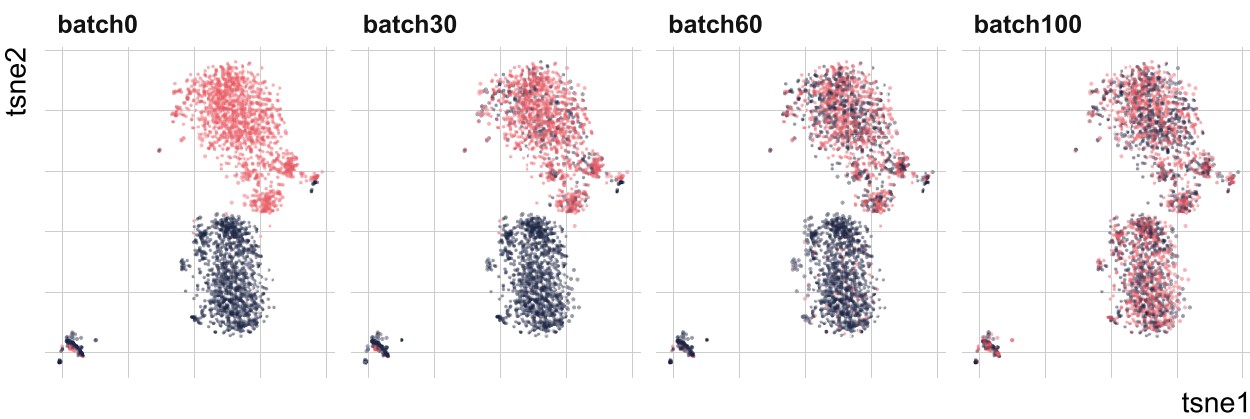

## Gradual increased randomness

patient ● pat1  ● pat2

**B**

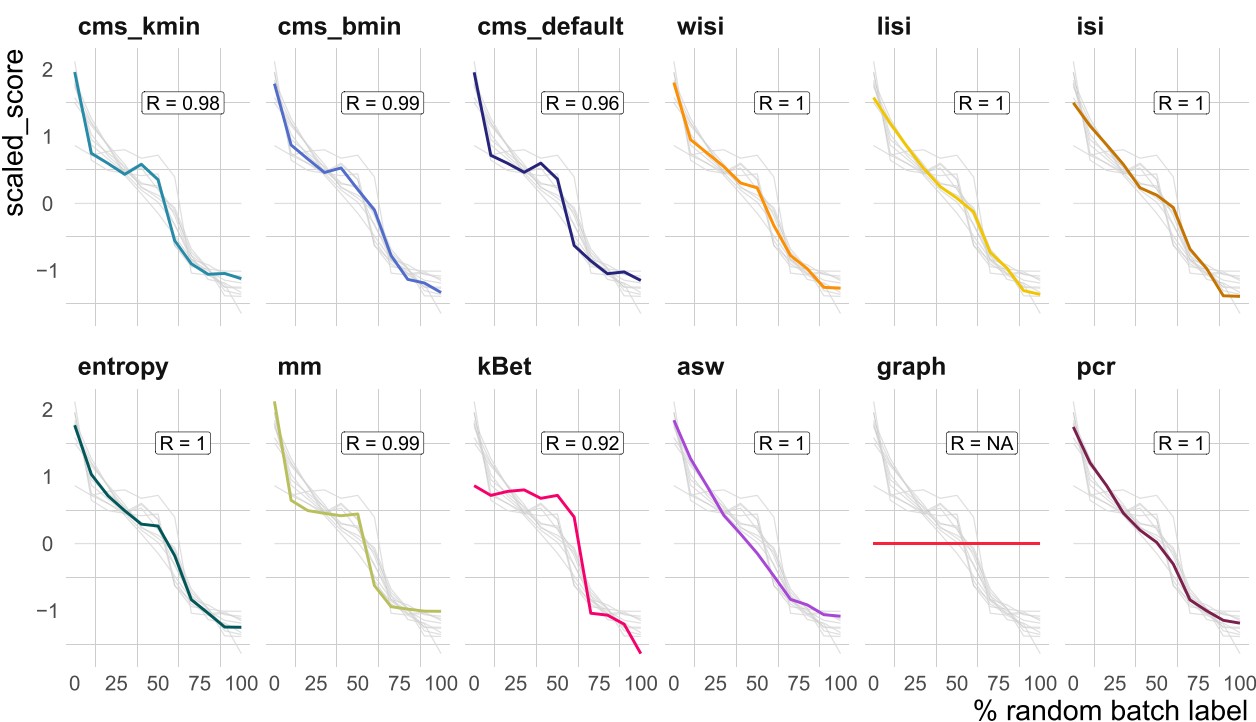

**Figure 4. Task 2—Scaling with batch label permutation.**
**(A)** A 2D tSNE projection of one semi-synthetic dataset. Batch labels were randomly permuted in 0–100% of the cells (batch 0 to batch 100) using steps of 10%. Expression profiles and cell type assignments remained unchanged. **(B)** Metric scores by increasingly randomized batch label in the dataset. Scores were standardized by subtraction of their mean and division of their SD across permutations. Directions were adjusted when necessary, such that all scores increase with batch strength. Grey lines indicate the scores of the other metrics. Corresponding absolute values of the Spearman correlation coefficients (R) are shown in the text box of each subpanel.

run times and memory usage according to the number of cells (see Fig 7). kBet took by far the longest CPU time (~38 h 30 min for 80,768 cells and 23,381 genes) and required the most memory, followed by cms_bmin (~6 h). graph, entropy, and lisi were the fastest and most memory efficient metrics (~10.8 min, ~11.4 min). cms_default and cms_kmin were the only metrics with CPU times independent of the number of genes. Although asw was among the fastest metrics, it failed due to memory limitations for the largest dataset.

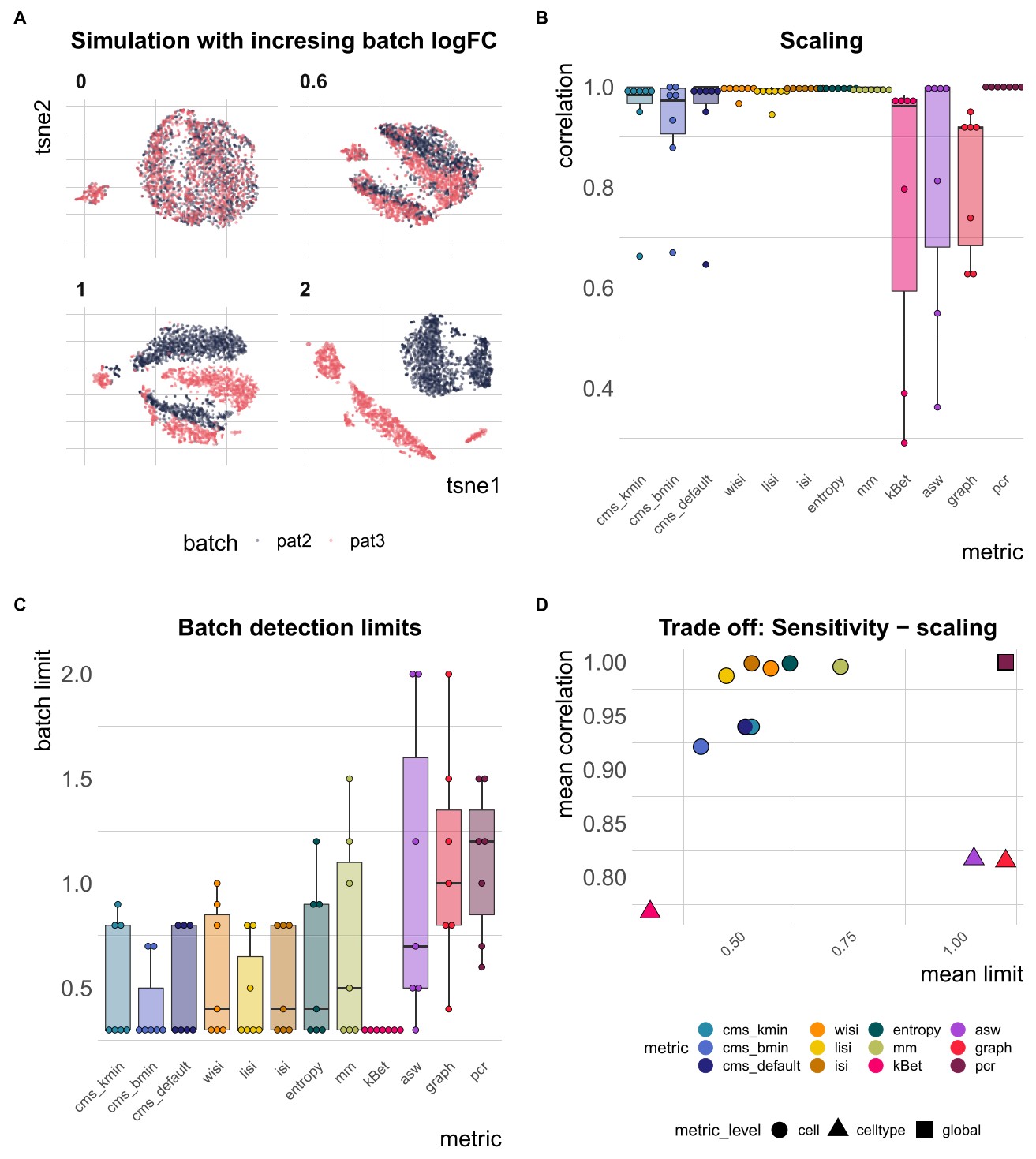

**Figure 5.   Task 3—Scaling and batch limits.**
**(A)** A 2D tSNE projection of simulation series with increasing batch logFC factors ($\theta_b$ from Equation (3); for example, a factor of 0 represents a batch-free dataset). **(B)** Boxplots of Spearman correlation coefficients of the metric scores and the relative batch strength for all seven simulation series. **(C)** Boxplots of batch limits, defined as the smallest batch logFC factors such that metrics differ more than 10% from the batch-free score. A small batch limit indicates high sensitivity to detect variation related to the batch variable. **(D)** Trade-off between batch detection sensitivity (batch limits) and scaling with batch strength. Shapes refer to metric types.

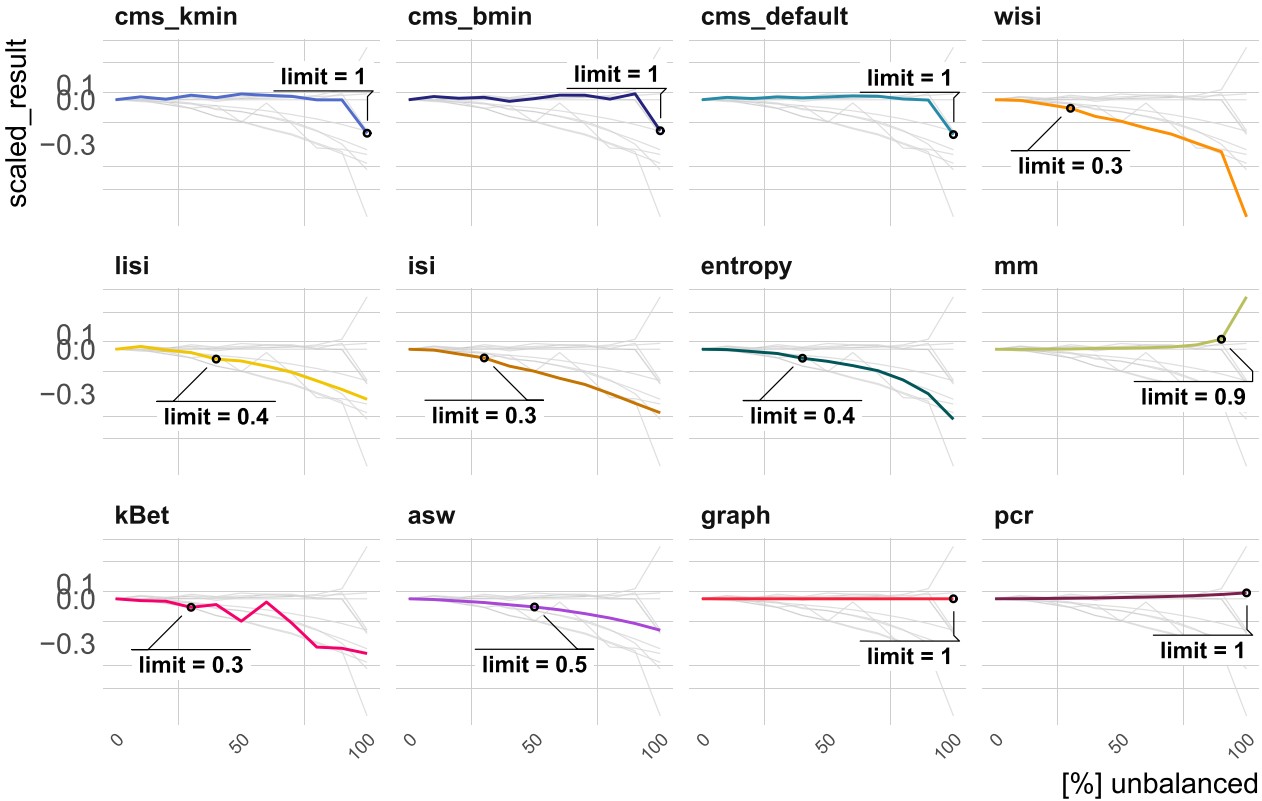

**A**

## Metric scores by increasingly imbalanced batch effects
### Simulated data with a moderate batch effect using csf_patient as reference.

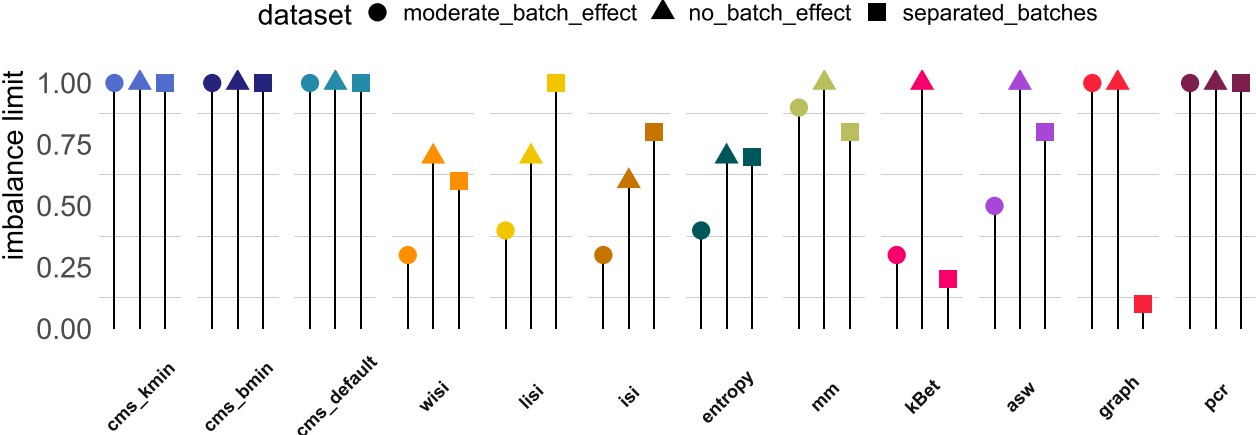

**B**

## Imbalance limits of metric score's divergence

**Figure 6. Task 4—Imbalance limits.**
**(A)** Changes in metric scores (within the same dataset) according to amount of imbalance. Imbalance refers to the increased removal of cells (0–100%) of one batch in one cell type. Scores were centred by the score with equal number of cells from both batches (no removed cells) and scaled by the score's range. Because gene-wise batch logFCs remain unchanged, metrics that are able to robustly quantify the batch effect, despite differential abundance, should show a stable score around 0. Grey lines indicate the other metrics' scores. **(B)** Lollipop plots of imbalance limits across different batch scenarios. Imbalance limits are defined as the proportion of removed cells from one batch and cell type that results in a metric score that differs more than 5% from the score of the balanced batch (no cell removed).

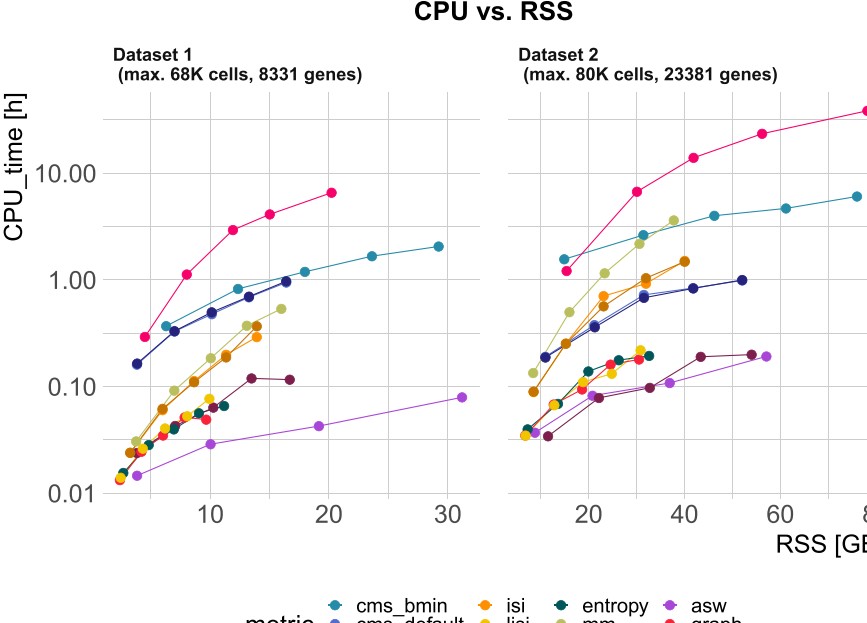

**CPU vs. RSS**

**Figure 7.   Task 5—Computational time and memory.** Metric's CPU time (logarithmic scale) versus maximum resident set size (RSS). Datasets were down-sampled in steps of 20% starting with 68,000 cells (Dataset 1) and 80,000 cells (Dataset 2). The two datasets also differ by the number of genes: Dataset 1 contains 8,331 genes and Dataset 2 contains 23,381 genes.

## Discussion/Summary

In this study, we explored batch effects and evaluated metrics to detect and quantify batch mixing in scRNA-seq data. We analysed batch characteristics in a variety of real datasets and observed different magnitudes of batch effects, different numbers of affected genes, and different levels of cell type specificity. Targeted towards these characteristics, we developed a metric, cms, to quantify batch mixing in scRNA-seq data, and consolidated the wide range of metrics to quantify batch effects within the CellMixS R/Bioconductor package (14).

We compared the performance of cms against existing metrics in a broad benchmark; in particular, we designed several tasks to assess: (i) whether a metric scales with batch strength in an inter- and intra-dataset comparison; (ii) whether a metric scales with increasingly random batch labels; (iii) a metric's sensitivity to detect batch effects; (iv) whether a metric is stable towards differential abundance of cell types; (v) computational cost. To evaluate performance, we generated synthetic data by adjusting the muscat simulation framework (10) to reflect batch characteristics of real data, using cell type–specific batch logFCs for each gene. An overall summary of performance is shown in Fig 8, using thresholds to categorize metrics into good, intermediate, or poor performance. Overall, we find considerable differences between metrics that can explain previously observed discrepancies (4 Preprint, 5). Since Task 1 was based on real data, there is no ground truth. Here, we estimated PVE-Batch and the proportion of DE genes as surrogates to evaluate metric performance (and applied less stringent thresholds for categorization). We used more stringent thresholds for tasks based on ground truth within synthetic data (Task 3 and Task 4). Except for graph, all methods pass the negative control test (i.e., scale with increasing batch label permutation). Because graph is a measure for the connectivity of cells from the same cell type, it

*indirectly* infers the strength of the batch effect as a relative distortion within cell type graph connections. Thus, graph is independent of the batch labels themselves and cannot be evaluated in Task 2. In summary, all cell-specific methods (cms, lisi, entropy, and mm) perform well in most tasks. pcr, the only global method tested, shows a stable performance in most tasks, but low sensitivity to detect mild batch effects in synthetic data (Task 3). In contrast, cell type–specific methods (asw, kBet, and graph) show a low performance across several tasks, for example, they fail to scale with an increased batch effect.

Cell- and cell type–specific metrics cannot differentiate between strong and very strong batch effects, as they become saturated at their maximal score in the simulation series of increasing batch effects (Task 3). They are designed to find local patterns and only consider neighbours for batch mixing; thus, they cannot detect global shifts of these neighbourhoods. One could partly address this by increasing the neighbourhood size, but this somewhat defeats the purpose of using cell-wise scores. Last, large differences in computation and memory demands were observed. lisi, pcr, and entropy should be considered when computational resources are limiting; for datasets with more than 50,000 cells, kBet, and cms_bmin cannot be recommended.

We tested different parameters that weight the neighbourhoods for cms and lisi. In the across-dataset comparison, no neighbourhood weighting (isi) and neighbourhoods determined by a minimal number of cells per batch (cms_bmin) showed the lowest correlation with batch strength characteristics. Altogether, we find that weighted metrics (lisi and wisi) outperform unweighted metrics (isi) and neighbourhoods with a fixed number of nearest neighbours (cms_default) or dynamic neighbourhoods (cms_kmin) outperform neighbourhoods determined by a minimal number of cells per batch (cms_bmin), whereas reducing the computational costs.

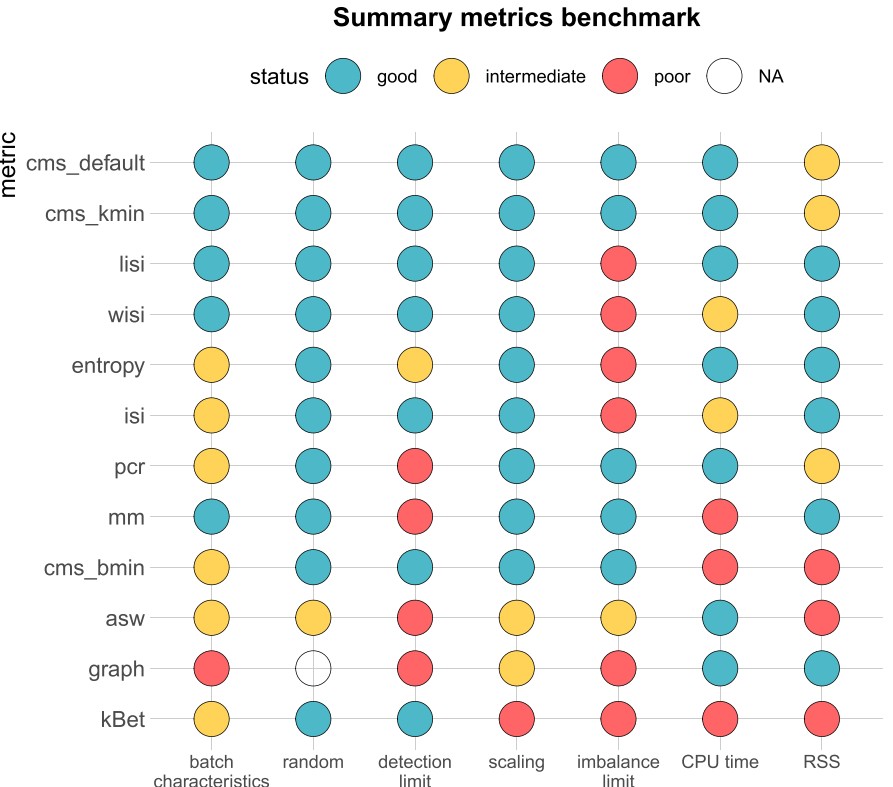

**Summary metrics benchmark**

**Figure 8.   Benchmark summary.**
Metric performance and ranking by benchmark task. Methods are ranked by their overall performance from top to bottom (numerical encoding good = 2, intermediate = 1, poor/NA = 0). Following thresholds for good and intermediate were used: batch characteristics and random: Spearman correlation coefficients ≥0.75, ≥0.5; scaling: Spearman correlation coefficient ≥0.9, ≥0.8; detection limits of ≤0.6, ≤0.7; imbalance limits ≥0.9, ≥0.75; CPU time ≤1, ≤2 h; RSS ≤50 GB, ≤70 GB.

Although most of the metrics have a clear interpretation, some (e.g., mm and graph) are less intuitive. Because mm's range changes with the number of batches, it is not suitable for across-dataset comparisons; instead, mm is most useful in a comparative setting (e.g., before and after integration) within the same dataset. Similarly, cell type composition and cell state changes affect graph connectivity; therefore, interpretations of the absolute value of graph are cell type– and dataset-specific. In general, cell-specific scores provide additional information about locality and can still be aggregated at the cell type or global level. This additional information can be useful to detect and understand local differences of a batch effect and guide strategies to account for it in downstream analysis. While cell type–specific metrics also provide local information, they depend on clustering and cell type assignment, which themselves can be affected by the batch effect; thus, it is desirable to have batch effect assessments that are independent of cell type assignment. If cell type information exists, cell-specific metrics can be run independently for each pre-determined cell type to assess interference of batch and cell type effects.

Taking all our results together, cms and lisi are the top performing methods for quantifying batch effects. They both show a consistently high performance across all tasks, except that lisi exhibits less stability toward cell type composition imbalance and cms has higher CPU and memory requirements.

The various cell-specific metrics are implemented in the CellMixS R/Bioconductor package and all code for analyses here are available at https://github.com/almutlue/batch_snakemake and https://github.com/almutlue/mixing_benchmark (DOI:10.5281/zenodo.4312672). Although CellMixS was developed with a focus on scRNA-seq

data, the implemented metrics are suited to a variety of genomic data types, such as multiomics or high-dimensional cytometry data. Furthermore, CellMixS uses established Bioconductor infrastructure and can, therefore, be easily applied within other workflows.

# Materials and Methods

### Batch characterization

We used the fitVarPartModel function of the variancePartition R package (11) to, for each gene, fit a linear mixed model to partition the variance attributable to batch, cell type, and the interaction of batch and cell type, as described in Equation (1). As a second measure, we used the number of significant DE genes between batches. DE genes were derived for each cell type using the edgeR R package (15). As described in reference 16, we treated single cells as pseudo-replicates using glmQLFTest to determine DE genes with an adjusted $P$-value ≤ 0.05. The derived cell type–specific batch logFCs (for each gene) were used to evaluate cell type specificity and generate synthetic datasets (see the Simulation section).

### Simulation

We adjusted the muscat (10) simulation framework for scRNA-seq data to include batch effects with characteristics of real data. As in the original framework, we estimated parameters from a reference

dataset and final counts were sampled from a (non-zero–inflated) negative binomial distribution, as described in Equation (2). Batch and cell-specific means, $\mu_{gcb}$, were derived as described in Equation (3). A batch tuning factor, $\theta_b$, was included to modulate the batch strength by multiplying the cell type–specific logFCs. Thus, gene and cell type correlations of the batch effect vector were propagated from the reference dataset to the synthetic data. We compared features between reference and simulated data using the countsimQC (17) R package and compared batch characteristics as PVE-Batch, PVE-Celltype, PVE-Int (see Fig S6A and B), and cell type–specific logFC distributions between reference and simulated datasets (see Fig S7A–F). Detailed analysis of each reference and corresponding simulations are provided at https://almutlue.github.io/batch_snakemake/index.html.

## Benchmark

We compared metrics used to quantify batch effects in scRNA-seq data in five benchmark tasks (see Table 2). For each task, we set a threshold to determine good, intermediate, or poor performance of each metric (see Fig 8). Task 1 and Task 2 received a threshold of Spearman correlation coefficients ≥0.75 for good and ≥0.5 for intermediate. We used more stringent thresholds for tasks based on synthetic data: Spearman correlation coefficient ≥0.9 and ≥0.8 for Task 3 scaling, average detection limits ≤0.6 and ≤0.7 for Task 3 detection limits and imbalance limits ≥0.9 and ≥0.75 for Task 4. For Task 5, we used the largest reference dataset (80,768 cells and 23,381 genes). We used a CPU time ≤1 and ≤2 h on this dataset as thresholds for time and a RSS ≤50 and ≤70 GB as thresholds for memory usage. To compare metrics, scores from cell-specific and cell type–specific metrics were aggregated to the global level. We used the mean score across all cells as aggregated score, thus to aggregate cell type–specific metrics, we used the mean (weighted by the number of cells).

## Metrics

### Cell-specific mixing score
The cell-specific mixing score compares batch-wise distance distributions according to each cell's knn. Euclidean distances within principal component space (top 10 components) are used. The k-sample test uses the Anderson–Darling criterion to test the hypothesis that k independent samples with sample sizes $n_1,...,n_k$ arose from a (potentially unspecified) common continuous distribution $F$. It is a rank test, and here the asymptotic $P$-value approximation from the kSamples (18) R package is used. cms is implemented in the CellMixS (14) R package and here we used $k$ = 200 nearest neighbours and three different ways of neighbourhood specification. In default mode (cms_default), all $k$ neighbours are included into the distance distributions. To determine cms_kmin, only cells before the first local minimum in the overall distance distribution of neighbouring cells were included. This assumes that the local minimum in the overall distance distribution defines a less dense region of neighbouring cells, corresponding to a (biologically) meaningful cell separation, for example, another cell type cluster. We used $k_{min}$ = 80 as a minimal number of cells to include.

cms_bmin was computed by including at least $b_{min}$ = 80 cells per batch.

### Lisi
The inverse Simpson's index was proposed to calculate the diversity within each cell's neighbourhood in scRNA-seq data (19). It represents the probability that different entities (here batches) are taken at random within a specified neighbourhood and its inverse represent the effective number of these entities. For each cell, the probability of a certain batch, $p(b)$, is determined from the batch abundances within its knn. The inverse of the sum of probabilities from all batches $B$ is defined as inverse Simpson index:

$$isi = \frac{1}{\sum_{b=1}^{B} p(b)}. \tag{4}$$

The local inverse Simpson index uses Gaussian kernel–based distributions of neighbourhoods for distance-based neighbourhood weighting to be sensitive towards local batch diversification within the knns. We used the lisi (19) R package with default parameters to determine the lisi score; CellMixS (14) to determine the isi and wisi score with $k$ = 200. The wisi score uses a simplified way of distance-based neighbour weighting by $\frac{1}{d^2}$, with $d$ representing the Euclidean distance in principal component space.

### Mixing metric
The mm uses the median position of the k-th cell ($k_{pos}$) from each batch within its knns as a score to quantify batch mixing (13). In general, the lower the score, the better mixed the neighbourhood is, but its absolute range depends on the number of batches and $k_{pos}$, for example, even with constant $k_{pos}$ (default = 5) and constant size of knn (default = 300), the metric's minimum is 10 for a dataset with three batches and 15 for a dataset with five batches. By default, CellMixS uses $k$ = 300 and $k_{pos}$ = 5 to compute the mm.

### Shannon's entropy
Shannon's entropy can be interpreted as the level of information in a random variable and thus also used as a measure for the level of information of the batch variable in a scRNA-seq dataset (5, 6 Preprint). We calculated the Shannon's entropy of the batch variable within each cell's knn using the relative abundance of batch $i$ as probability, $p(x_i)$, for batch $i$ across all batches $n$:

$$entropy = -\frac{1}{|n|} \sum_{i=1}^{n} p(x_i) * \log(p(x_i)). \tag{5}$$

The entropy is divided by the number of batches $|n|$ to get a range between 0 and 1 with 0 corresponding to a low level of randomness in the data. Here, we use $k$ = 200 to calculate the entropy for each cell.

### k-nearest-neighbour batch effect test (kBet)
The knn batch effect test was specifically developed to detect batch effects in scRNA-seq data (2). It compares the batch label composition of random cell neighbourhoods (local) with the overall (global) batch label composition using a $\chi^2$-test. This is repeated for a random subset of cells and summarized in a rejection rate of the

**Life Science Alliance**

hypothesis (i.e., local and global batch compositions are the same). We computed kBet separately for each cell type to account for differences in the batch compositions between cell types.

### Graph connectivity

Graph connectivity has been proposed as a metric to evaluate batch integration methods based on within-cell type connections (4 *Preprint*). It determines the relative connectivity of cells from the same cell type in the knn graph *G*. graph relies on the assumption that in a batch-free dataset, all cells from the same cell type *t* are directly connected within the subset graph $G(N_t, E_t)$ (with nodes $N_t$ and edges $E_t$). The score represents the average proportion of the directly connected part (longest connected component: *LCC*) of the cell type graph relative to the overall number of cells per cell type $N_t$ across all cell types *T*:

$$graph = \frac{1}{|T|} \sum_{t \in T} \frac{|LCC(G(N_t, E_t))|}{N_t}. \quad (6)$$

The graph ranges from 0 to 1, with 1 corresponding to completely connected subset graphs and thus a batch-free dataset. We used the buildKNNGraph function of the scran R package (20) with $k = 5$ to generate the knn graph for each dataset and determined the number and length of connected components using the components function of the igraph (21) R package.

### ASW

The asw is an established measure for cluster stability as it describes the relation of within-cluster distances/similarities and between-cluster distances/similarities. To apply asw on batch quantification, it is used to assess the stability of batch clusters (2, 4 *Preprint*, 5). Thus, asw is calculated to compare within-batch distances to between-batch distances. The asw of cell *i* in batch *j* is based on the average distances $a_i$ of cell *i* to all other cells from batch *j* and the average distances $d(i, B)$ to all other batches *B*. The difference between $a_i$ and the minimum $b_i = min(d(i, B))$ of these average distances describes the within-batch relation of cells compared with the closest other batch. The silhouette width of cell *i* is defined as follows:

$$s_i = \frac{a_i - b_i}{max(a_i, b_i)}. \quad (7)$$

We used the average of all silhouette widths from the same cell type as asw score. The silhouette width was computed using the silhouette function of the R package cluster (22) with Euclidean distances within the top 10 principal components of PCA space. Here we report $asw_{batch} = 1 - abs(asw)$ as asw score with a range between 0 and 1, with 0 corresponding to completely separated batches and 1 to mixed batches.

### PCR

PCR (2, 4 *Preprint*) is based on PCA of the log normalized count matrix *C*. For each principal component $PC = \{pc_1, ..., pc_N\}$, the coefficient of determination $R^2$ from a linear regression of the batch variable *B* is calculated to get $R^2(PC|B)$. These coefficients are weighted by the variance related to the principal components $Var(C|PC)$. The pcr score represents the overall contribution of the batch variable to the variance of a dataset and is determined by the sum of all weighted coefficients of determination:

$$pcr = \sum_{i=1}^{N} Var(C|PC_i) * R^2(PC_i|B). \quad (8)$$

We used the pca function of the PCAtools R package (23) on logcounts of the 1,000 most variable genes to get principle components and their corresponding variance. We then used the base R cor function to get the Pearson's correlation coefficient of each PC and the batch variable. Its square is equivalent to the coefficient of determination from a linear regression of the variables. We calculated pcr as described above and scaled it by the overall variance represented of the top 100 PCs.

### Datasets

In total, seven datasets (some with multiple sources of batch effect) (see Table S1) were used. We used these datasets, simulations with these data as reference and semi-synthetic data (e.g., with permuted batch labels) to benchmark the metrics. All datasets were processed in a consistent way. For datasets where raw read counts were used as the starting point (pbmc2_pat/pbmc2_media, csf_pat/csf_media, pbmc_roche), we used the scDblFinder (24) R package to filter for doublet cells. Quality control, filtering and normalization was performed using the scater (25) and scran (20) R packages. Genes that were detected in less than 20 cells and cells with feature counts, number of expressed features, and percentage of mitochondrial genes beyond 2.5 Median Absolute Deviations in either direction from the median were excluded from further analysis. All datasets were normalized by first scaling counts cell-wise using pool-based size factors from library sizes (26) and then using log transformation with a pseudocount of one. In datasets without existing cell type assignments (pbmc2_pat/pbmc2_media, csf_pat/csf_media, pbmc_roche), Seurat (13) R package v.3.0 was used for integration based on canonical correlation analysis and integration anchors from mutual nearest neighbours (1). Integrated cells were clustered with a resolution of 0.2, including the 500 most HVGs as identified by Seurat's FindVariableFeatures. All data processing scripts are publicly available at https://almutlue.github.io/batch_dataset/.

### Dataset 1: CellBench

The single cell mixology benchmark dataset (27) is provided in the CellBench R/Bioconductor (28) package. The dataset consists of cells from three human lung adenocarcinoma cell lines (HCC827, H1975, and H2228). Equally mixed samples of these cell lines have been processed and sequenced using three different protocols: CEL-seq2, Drop-seq (with Dolomite equipment) and 10X Chromium. This dataset has been generated for benchmarking purposes and provides a resource of 1,401 cells from three known batches attributed to different sequencing protocols with identical cell line compositions. Here, we treat the cell lines as analogues for cell types.

### Dataset 2: Human Cell Atlas—Mereu (hca)

This dataset has been generated to compare different sequencing protocols for scRNA-seq by Mereu et al (29), as part of the human

cell atlas project. The dataset consists of 20,237 cells, which are PBMCs and the human kidney cell line HEK293T. Equal proportions of these cells have been sequenced by 13 different sequencing protocols: C1HT-medium,C1HT-small, CEL-Seq2, Chromium, Chromium(sn), ddSEQ, Drop-Seq, ICELL8, inDrop, MARS-Seq, mcSCRB-Seq, Quartz-Seq2, and Smart-Seq2. We use cell type annotations as provided in the original analysis using the R package matchSCore2 (29).

### Dataset 3: PBMC media and patient (pbmc2_pat/pbmc2_media)

Fresh PBMCs were harvested using Ficoll–Paque density gradient centrifugation from 15 ml whole blood specimens of two unrelated donors obtained from Blutspendezentrum SRK beider, Basel. Cells were washed with 1× PBS, resuspended in 20 ml 1× PBS, and counted using trypan blue staining on Countess II (Life Technology). For each donor's PBMCs, three separate aliquots were prepared at 300 cells/μl and suspended in 300 μl of RPMI-1640 plus 10% FBS. Aliquot 1 (fresh sample): cells were centrifuged (10 min, 300$g$) and the pellet was resuspended in 80 μl PBS-0.04% BSA. Aliquot 2 (MetOH fixation and cryopreservation): prechilled MetOH (four volumes) was added dropwise before cells were incubated (30 min, –20°C) and stored at –80°C for 7 d. After cryopreservation, cells were thawed, equilibrated, and centrifuged (10 min, 300$g$) at 4°C. The cell pellet was resuspended in 50 μl SSC cocktail (0.04% BSA, 1% RNAse inhibitor, and 40 mM DTT in SSC 1×). Aliquot 3 (cryopreservation in 15% DMSO): cells were centrifuged (10 min, 300$g$), resuspended in 500 μl RPMI-1640 plus 40% FBS, and diluted using 500 μl prechilled freezing medium (RPMI-1640 plus 40% FBS and 30% DMSO). The sample was placed inside a freezing container and stored at –80°C overnight before moving it to a liquid nitrogen tank for 7 d storage. After cryopreservation, the cells were thawed in a water bath (37°C), diluted 1:4 by RPMI-1640 plus 10% FBS, and centrifuged (10 min, 300$g$). The cell pellet was resuspended using 50 μl RPMI-1640 plus 10% FBS.

For all six samples (2 patients × 3 aliquots), the cells were counted using trypan blue staining on Countess II (Life technology), and a total of 12,000 cells/sample were loaded on the 10× Single Cell B Chip. cDNA libraries were prepared using the Chromium Single Cell 3' Library and Gel Bead kit v3 (10× Genomics) according to the manufacturer's instructions. Libraries were sequenced using Illumina Hiseq 4000 using the HiSeq 3000/4000 SBS kit (Illumina) and HiSeq 3000/4000 PE cluster kit with a target sequencing depth of 30,000 reads/cell.

### Dataset 4: cerebrospinal fluid (csf_pat/csf_media)

3 ml of cerebrospinal fluid (CSF) per patient was collected from the diagnostic lumbar puncture of three patients at the University Hospital Basel. All patients consented to CSF draw and the procedure was performed according to University Hospital IRB guidelines. Each CSF sample was centrifuged (10 min, 400$g$) and resuspended in 60 μl RPMI-1640 plus 40% FBS. Cells were counted using trypan blue staining on Countess II (Life Technology) and processed as following: Sample 1 (fresh CSF): freshly isolated cells were counted, assessed for cell viability, and loaded on the 10× Single Cell B Chip. Samples 2 and 3 (cryopreserved CSF): cells were suspended in 450 μl of RPMI-1640 plus 40% FBS and diluted using 500 μl prechilled freezing medium (RPMI-1640 plus 40% FBS and

30% DMSO). Before moving samples to a liquid nitrogen tank, they were placed inside a freezing container and stored at –80°C overnight. After 7 d, the samples were thawed in a water bath (37C), diluted 1:4 using RPMI-1640 plus 10% FBS, and centrifuged (10 min, 300$g$). The cell pellet was resuspended with 50 μl RPMI-1640 plus 10% FBS, cells were counted using trypan blue staining on Countess II (Life Technology), and samples with at least 200 cells/μl and a cell viability above 60% were loaded on a 10× Single Cell B Chip. cDNA libraries preparation and sequencing was performed as described for Dataset 3.

### Dataset 5: PBMC Roche

Fresh PBMCs were isolated from whole blood specimens of four donors following the protocol described for Dataset 3. For each donor PBMCs, four separate aliquots were prepared at 500 cells/μl and suspended in 200 μl of RPMI-1640 plus 10% FBS. Aliquot 1 (fresh sample) and Aliquot 2 (cryopreservation in 15%DMSO): Cells were prepared as described in Dataset 3. Aliquot 3 (cryopreservation in CS10 media) and Aliquot 4 (cryopreservation in PSC media): Cells were centrifuged (10 min, 300$g$) and resuspended by drop-wise addition of 1 ml prechilled CryoStor CS10 medium (StemCell Technologies) for Aliquot 3 and 1 ml prechilled PSC (Thermo Fisher Scientific) for Aliquot 4. Cells were placed inside a freezing container and stored at –80°C overnight before moving them to a liquid nitrogen tank for 7 d storage. After cryopreservation samples were thawed in a water bath (37°C), diluted 1:4 using RPMI-1640 plus 10% FBS (Aliquot 4: plus 0.01% RevitaCell [Thermo Fisher Scientific]), and centrifuged (10 min, 300$g$). The cell pellet was resuspended in 50 μl RPMI-1640 plus 10% FBS. For all samples, the cells were counted using trypan blue staining on Countess II (Life Technology), and a total of 12,000 estimated cells from each sample were loaded on a 10× Single Cell B Chip. cDNA library preparation and sequencing were performed as described for Dataset 3 (except v2 of the Gel Bead kit was used).

### Dataset 6: Kang

The Kang dataset consist of 10× droplet-based scRNA-seq PBMC data from eight Lupus patients before and after 6 h of treatment with INF-$β$ (30). Here, we limit our analysis to the untreated control data resulting in 14,619 cells from eight different patients. Data and cell type annotations were accessed via the muscData (31) R/Bioconductor ExperimentHub (32) package.

### Dataset 7: pancreas

The pancreas dataset is a collection of human pancreatic islet cell datasets produced across three technologies: CelSeq, CelSeq2, and SMART-Seq2. In total, there are 5,683 cells. Data and annotations were accessed via the SeuratData (33) R package.

## Software specifications and code availability

If not stated otherwise, analyses were run in R v3.6 (34) using Bioconductor v3.10 (35). R packages ggplot2 (36) and ComplexHeatmap (37) were used to visualize results. Package versions of all used software are summarized in Supplemental Data 1. Code and a browseable workflow (38) for data pre-processing can be found at https://almutlue.github.io/batch_dataset/ (DOI:10.5281/zenodo.4312591).

Batch characterization and simulation was run as a snakemake (39) pipeline. Code and deployment of results are available at https://github.com/almutlue/batch_snakemake (DOI:10.5281/zenodo.4312603) and https://almutlue.github.io/batch_snakemake/index.html. The deployment also includes reports with count matrix comparisons between real and synthetic data generated by countsimQC (17). Cell-specific metrics are made available in the open-source CellMixS R/Bioconductor package (14). CellMixS includes a different way of neighbourhood weighting for the lisi score implemented as wisi and isi scores. Lisi itself was run using the lisi R package (19). kBet was run as implemented in the kBet R package (2). Code used to run the remaining metrics, namely pcr, graph and asw is available at https://github.com/almutlue/mixing_benchmark (DOI:10.5281/zenodo.4312672). This also includes the code to run the metrics benchmark. Results and further descriptions are shown at https://almutlue.github.io/mixing_benchmark/index.html.

## Data Availability

Data accession numbers for all data are provided in Table S1. Data objects including datasets in SingleCellExperiment format are available from DOI:10.6084/m9.figshare.13341200. Supplemental Data 1 is available from DOI:10.6084/m9.figshare.13341200.

## Supplementary Information

## Acknowledgements

We thank Mechthild Lütge, Pierre-Luc Germain, and members of the Robinson Lab at the University of Zurich for their valuable feedback and input. This work was supported by the Swiss National Science Foundation (grant number: CRSII5_177208). MD Robinson acknowledges the support from the University Research Priority Program Evolution in Action at the University of Zurich.

### Author's Contributions

A Lutge: conceptualization, data curation, software, formal analysis, visualization, methodology, and writing—original draft, review, and editing.
J Zyprych-Walczak: data curation, formal analysis, and methodology.
U Brykczynska Kunzmann: data curation, formal analysis, and methodology.
HL Crowell: software and formal analysis.
D Calini: data curation and investigation.
D Malhotra: supervision, funding acquisition, investigation, and project administration.
C Soneson: conceptualization, supervision, investigation, methodology, and writing—original draft, review, and editing.
MD Robinson: conceptualization, supervision, funding acquisition, investigation, methodology, project administration, and writing—original draft, review, and editing.

### Conflict of Interest Statement

D Malhotra is a full-time employee of Roche; the remaining authors declare that they have no competing interests.

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
