## [Reviewer comments · Life Science Alliance]

Life Science Alliance

CellMixS: quantifying and visualizing batch effects in single cell RNA-seq data

Almut Lutge, Joanna Zypych-Walczak, Urszula Brykczynska Kunzmann, Helena Crowell, Daniela Calini, Dheeraj Malhotra, Charlotte Soneson, and Mark Robinson

DOI: [https://10.26508/lsa.202001004](https://doi.org/10.26508/lsa.202001004)

Corresponding author(s): Mark Robinson, University of Zurich and Almut Lutge, University of Zurich, Departement of Molecular Life Sciences

Review Timeline:

Submission Date:	2020-12-21
Editorial Decision:	2021-01-20
Revision Received:	2021-02-02
Editorial Decision:	2021-02-18
Revision Received:	2021-03-09
Accepted:	2021-03-09

Scientific Editor: Shachi Bhatt

Transaction Report:

January 20, 2021

Re: Life Science Alliance manuscript #LSA-2020-01004-T

Prof. Mark D Robinson
University of Zurich
Institute of Molecular Life Sciences
Winterthurerstrasse 190
IMLS
Zurich, ZH 8057
Switzerland

Dear Dr. Robinson,

Thank you for submitting your manuscript entitled "CellMixS: quantifying and visualizing batch effects in single cell RNA-seq data" to Life Science Alliance. The manuscript was assessed by expert reviewers, whose comments are appended to this letter.

As you will note from the reviewers' comments, both reviewers were quite enthusiastic about these data, but reviewer 1 has mentioned a number of concerns and questions that should be addressed prior to further consideration of the manuscript at LSA. We, thus, encourage you to submit a revised version of the manuscript that addresses all of the reviewers' concerns.

Thank you for this interesting contribution to Life Science Alliance. We are looking forward to receiving your revised manuscript.

Sincerely,

Shachi Bhatt, Ph.D.
Executive Editor
Life Science Alliance
<https://www.lsjournal.org/>
Tweet @SciBhatt @LSAJournal

- A letter addressing the reviewers' comments point by point.
- An editable version of the final text (.DOC or .DOCX) is needed for copyediting (no PDFs).
- High-resolution figure, supplementary figure and video files uploaded as individual files: See our detailed guidelines for preparing your production-ready images, <https://www.life-science-alliance.org/authors>
- Summary blurb (enter in submission system): A short text summarizing in a single sentence the study (max. 200 characters including spaces). This text is used in conjunction with the titles of papers, hence should be informative and complementary to the title and running title. It should describe the context and significance of the findings for a general readership; it should be written in the present tense and refer to the work in the third person. Author names should not be mentioned.

B. MANUSCRIPT ORGANIZATION AND FORMATTING:

Reviewer #1 (Comments to the Authors (Required)):

The manuscript presented by Lutge and colleagues describes a new metric, cell-specific mixing score (csm), designed to detect and quantify batch effects in scRNA-seq data. The metric is based on the Anderson-Darling test, which is used to test the null hypothesis of "no batch effect". In single-cell data analysis, it is important to detect batch effects and there are already several methods designed for this task. In some cases, the idea behind these methods is close to the one proposed here. However, it is probably true that a systematic comparison of batch mixing metrics in

key tasks is relevant and has not been conducted yet.

Overall, this study is well constructed and presented, the manuscript well written and easy to follow. Real and simulated datasets are used to evaluate the csm method in distinct scenarios.

Here my comments:

Authors evaluated the variance contribution of batch effects related to distinct sources, such as batch, cell type and interaction between them. In figure 1, they observed that most of the gene variances are attributed to batches and cell types and a lower percentage is related to their interaction. Given that, how can the authors motivate the fact that csm does not account for cell type assignment? Did authors evaluate if the cell type identity of the cell in the neighbourhood can affect the csm metric?

If distances are derived from the PCA space, the structure of the data is partially retrieved. In this way, the cells of the same type will be closer if we consider a scenario with no batch effect at a specific neighbourhood. In contrast, if a neighbourhood is affected by cell type-specific batch effects, cells of random cell type composition could be in that neighbourhood even if they are from distinct batches. Have authors considered this aspect?

About the comparison of metric scores for the task 1, authors say: "Most cell-specific metrics showed a plateau in their score towards higher batch strength, suggesting a maximal score has been reached and thus they cannot further discern the strength of a batch effect". I believe this is not optimal. However, authors have not commented on that. In principle, if the proportion of DE genes is very high and a plateau is observed, this is desirable. However, here the plateau seems to start quite early (~15%). This would mean that any of these metrics is able to reflect the real strength of the batch effect. I think this should be clarified. I understand that probably it is more important having high sensitivity for detecting batch effects. Authors should address this point and help the reading of these graphics.

In task 3 authors evaluate the sensitivity in detecting batch effects and, similarly to task 1, they also compare the ability of each metric to reflect the strength of the batch effect by increasing the batch logFC. Are these features equally important in a real context? Could the authors address this point. What is the impact of the scaling in a real scenario?

Are these tasks independent? Does task 3 (scaling) reflect task 1?

Minor comments:

The links to Table 1 (row 32), Figure S1 and Table S1 (row 56), Figure S2 (row 74) don't work for me. Plots in Figures 1B and 1C require more explanations. I find it difficult to understand the meaning of the dotted lines and their corresponding percentage values.

In figure 3, there are many annotations at each tool-specific plot and this makes their understanding difficult. Probably, the lack of both x-axes does not help.

Reviewer #2 (Comments to the Authors (Required)):

The authors discuss metrics / measures by which to assess / quantify the degree of batch effects affecting single cell RNA-seq experiments.

The authors suggest a new metric themselves. They also compare existing measures. They do this to a degree that leaves no further questions open.

I therefore consider the paper a great guideline when trying to get control of batch effects that

affect different runs of scRNA-seq experiments.

I only have (very) minor comments, and recommend to accept this paper.

MINOR:

Results:

* Figure 1A: would be preferable to have hca, cellbench, pancreas in one row, and so on

Discussion:

* Citation Crowell2019a broken

CellMixS reviewer's comments

Reviewer #1 (Comments to the Authors (Required)):

The manuscript presented by Lutge and colleagues describes a new metric, cell-specific mixing score (csm), designed to detect and quantify batch effects in scRNA-seq data. The metric is based on the Anderson-Darling test, which is used to test the null hypothesis of "no batch effect".

In single-cell data analysis, it is important to detect batch effects and there are already several methods designed for this task. In some cases, the idea behind these methods is close to the one proposed here. However, it is probably true that a systematic comparison of batch mixing metrics in key tasks is relevant and has not been conducted yet.

Overall, this study is well constructed and presented, the manuscript well written and easy to follow. Real and simulated datasets are used to evaluate the csm method in distinct scenarios.

Here my comments:

Authors evaluated the variance contribution of batch effects related to distinct sources, such as batch, cell type and interaction between them. In figure 1, they observed that most of the gene variances are attributed to batches and cell types and a lower percentage is related to their interaction. Given that, how can the authors motivate the fact that csm does not account for cell type assignment? Did authors evaluate if the cell type identity of the cell in the neighbourhood can affect the csm metric?

If distances are derived from the PCA space, the structure of the data is partially retrieved. In this way, the cells of the same type will be closer if we consider a scenario with no batch effect at a specific neighbourhood. In contrast, if a neighbourhood is affected by cell type-specific batch effects, cells of random cell type composition could be in that neighbourhood even if they are from distinct batches. Have authors considered this aspect?

We developed cms with the goal to provide a metric that was independent of any cell type assignment, motivated by the fact that cell type assignment can be affected by

the batch effect itself. So, cms is cell type independent to prevent bias related to the method of cell type assignment or possibly misclassified cells. For example, a cell type dependent metric could give different results on the same data depending on the clustering method and parameters used.

As pointed out by the reviewer, a cell type-specific batch effect could lead to mixing of cells from different cell types and batches, which would not be detected by a metric without considering cell type labels. To address this concern we added the following sentence (bold) to the Discussion section:

*While cell type-specific metrics also provide local information, they depend on clustering and cell type assignment, which themselves can be affected by the batch effect; thus, it is desirable to have batch effect assessments that are independent of cell type assignment. **If cell type information exists, cell-specific metrics can also be run independently for each pre-determined cell type to assess interference of batch and cell type effects.***

In Task 1, we tested metrics on datasets with batch effects varying in the cell type-specificity of the batch effect. Based on these results, we did not observe an advantage to including cell type information in the metric.

About the comparison of metric scores for the task 1, authors say: "Most cell-specific metrics showed a plateau in their score towards higher batch strength, suggesting a maximal score has been reached and thus they cannot further discern the strength of a batch effect". I believe this is not optimal. However, authors have not commented on that. In principle, if the proportion of DE genes is very high and a plateau is observed, this is desirable. However, here the plateau seems to start quite early (~15%). This would mean that any of these metrics is able to reflect the real strength of the batch effect. I think this should be clarified. I understand that probably it is more important having high sensitivity for detecting batch effects. Authors should address this point and help the reading of these graphics.

We agree that the plateau shown by cell-specific metrics in Task 1 is an important observation that should be clarified. We added the following paragraph (bold) to the Results section for Task 1 to give the reader more context for these results:

Most cell-specific metrics showed a plateau in their score towards higher batch strength, suggesting a maximal score has been reached and thus they cannot further discern the strength of a batch effect.

In Figure S2, we show 2D tSNE projections of all datasets ordered by their percentage of DE genes between batches. All datasets except the kang and pbmc_roche dataset exhibit clear batch effects that can easily be identified by visualization, where most neighbourhoods consist of cells from the same batch. While cell-specific metrics that only consider each cell's neighbourhood get saturated at their nominal minimum (from the csf_patient dataset onwards), their summarized score still reflects the overall order of datasets based on batch strength measures.

We also changed the caption of Figure 3 to help describe the corresponding graphic to the following (changes in bold):

*Task 1 - Reflection of batch characteristics: Metric scores versus (surrogate) batch strength across the real datasets. **Summarized metric scores (y-axis) are compared to the proportion of DE genes (top x-axis, solid line) and the mean PVE-Batch (bottom x-axis, dashed line) per dataset. Datasets with a stronger batch effect (high percentage of DE genes/mean PVE-Batch) are expected to show a higher overall metric score than datasets with mild batch effects (low percentage of DE genes/mean PVE-Batch).** Spearman correlation coefficients of metrics against the two batch strength measures are shown ($R_{PVE-Batch}$, R_{DE}) in the text boxes for each metric and evaluated in Task 1. Metric scores were standardized by subtraction of their minimum and division of their range (maximum - minimum) across datasets. Directions were adjusted when necessary, such that all scores increase with batch strength.*

In task 3 authors evaluate the sensitivity in detecting batch effects and, similarly to task 1, they also compare the ability of each metric to reflect the strength of the batch

effect by increasing the batch logFC. Are these features equally important in a real context? Could the authors address this point. What is the impact of the scaling in a real scenario?

Are these tasks independent? Does task 3 (scaling) reflect task 1?

We agree that a metrics ability to scale with the strength of a batch effect and its sensitivity are complementary aspects and both should be considered when interpreting a metric's result.

While it is desirable to have a sensitive metric that detects any bias related to a batch effect, not every structure related to the batch is a real confounder of the signal of interest. For example, a mild batch effect might confound the within cell type structure, but not cell type clusters themselves. Thus, the batch effect does not need to be considered for cell type assignment, but becomes relevant at the level of clustering to cell identity. As the relevance of a batch effect is context dependent, it is important for a metric to be sensitive, but also interpretable with regards to the severity of the batch effect. In Task 1, we evaluate whether the metrics reflect batch strength related characteristics *across* datasets and in Task 3, we evaluate the metrics ability to scale within the *same* dataset. The latter is particularly relevant in benchmarks for batch correction methods or when a batch effect before and after correction is compared.

To expand upon these considerations about sensitivity and scaling of metrics, we edited (bold) the following paragraphs in the manuscript:

Results section "Comparison of batch mixing metrics":

*Altogether, we designed 5 benchmark tasks to cover the most relevant use cases of these metrics (see Table 2 for short descriptions). **One major application of these metrics is to assess the severity of a batch effect and thus reflect the level of confounding.** For example, a larger score should result from a stronger batch effect across datasets (Task 1).*

Table 2, Task1, Aim:

*Test whether metrics reflect batch strength/**confounding** across datasets*

Results section “Task 1: Reflection of batch characteristics”:

*In this task, we tested a metric's ability to discriminate between a strong and a mild batch effect across datasets. **This is an important feature of these metrics as the impact of a batch effect is context-specific and depends on how strongly interesting data characteristics are confounded.** To test this, we used the batch characteristics and datasets explored above.*

Minor comments:

The links to Table 1 (row 32), Figure S1 and Table S1 (row 56), Figure S2 (row 74) don't work for me.

Thanks a lot for pointing this out. It seems to be related to the conversion of the .tex files. We will pay attention when uploading the improved version.

Plots in Figures 1B and 1C require more explanations. I find it difficult to understand the meaning of the dotted lines and their corresponding percentage values.

We changed the caption of Figure 1 to:

*... B,C) Batch logFC distribution by cell type and batch effect in the cellbench and hca datasets, respectively. Each column represents a density plot of the estimated logFCs for a batch / cell type combination. **Dotted lines indicate the mean, 25%, 50% and 75% percentiles.***

In figure 3, there are many annotations at each tool-specific plot and this makes their understanding difficult. Probably, the lack of both x-axes does not help.

We added axis lines and marks to Figure 3.

Reviewer #2 (Comments to the Authors (Required)):

The authors discuss metrics / measures by which to assess / quantify the degree of batch effects affecting single cell RNA-seq experiments.

The authors suggest a new metric themselves. They also compare existing measures. They do this to a degree that leaves no further questions open.

I therefore consider the paper a great guideline when trying to get control of batch effects that affect different runs of scRNA-seq experiments.

I only have (very) minor comments, and recommend to accept this paper.

MINOR:

Results:

* Figure 1A: would be preferable to have hca, cellbench, pancreas in one row, and so on

As suggested, we changed the order of datasets in Figure 1A to a more meaningful order with batches related to the same origin (sequencing protocols, patients, media storage) in the same row.

Discussion:

* Citation Crowell2019a broken

Thanks for pointing this out. We fixed the citation.

February 18, 2021

RE: Life Science Alliance Manuscript #LSA-2020-01004-TR

Prof. Mark D Robinson
University of Zurich
Institute of Molecular Life Sciences
Winterthurerstrasse 190
IMLS
Zurich, ZH 8057
Switzerland

Dear Dr. Robinson,

Thank you for submitting your revised manuscript entitled "CellMixS: quantifying and visualizing batch effects in single cell RNA-seq data". We would be happy to publish your paper in Life Science Alliance pending final revisions necessary to meet our formatting guidelines.

Along with the points listed below, please also attend to the following,

- please consult our manuscript preparation guidelines <https://www.life-science-alliance.org/manuscript-prep> and make sure your manuscript sections are in the correct order
- please make sure the author order in your manuscript and our system match, add all Contributing Authors in our system
- please add ORCID ID for secondary corresponding author-they should have received instructions on how to do so
- please add a Category for your manuscript in our system
- please upload your main and supplementary figures as single files
- please add callouts for Figures S4A,B,C,D,E,F; S6A, B and S7A,B,C,D,E,F to your main manuscript text
- please add your main, supplementary figure and table legends to the main manuscript text after the references section. Please make sure the manuscript sections are aligned in accordance with LSA's formatting guidelines: please separate the Figure legends and Supplemental Figure legends into separate sections
- please upload your Supplementary table in editable .doc or .xls files
- please upload your main manuscript text as an editable doc file
- we encourage you to revise the figure legend for figure S4 such that the figure panels are introduced in an alphabetical order

A. FINAL FILES:

B. MANUSCRIPT ORGANIZATION AND FORMATTING:

Thank you for this interesting contribution, we look forward to publishing your paper in Life Science

Alliance.

Sincerely,

Shachi Bhatt, Ph.D.

Executive Editor

Life Science Alliance

<https://www.lsjournal.org/>

Interested in an editorial career? EMBO Solutions is hiring a Scientific Editor to join the international Life Science Alliance team. Find out more here -

https://www.embo.org/documents/jobs/Vacancy_Notice_Scientific_editor_LSA.pdf

Reviewer #1 (Comments to the Authors (Required)):

The authors have done significant work to improve the manuscript. They answered all questions and addressed problematic points I commented last time. The manuscript is now of a very high quality.

March 9, 2021

RE: Life Science Alliance Manuscript #LSA-2020-01004-TRR

Prof. Mark D Robinson
University of Zurich
Institute of Molecular Life Sciences
Winterthurerstrasse 190
IMLS
Zurich, ZH 8057
Switzerland

Dear Dr. Robinson,

Thank you for submitting your Research Article entitled "CellMixS: quantifying and visualizing batch effects in single cell RNA-seq data". It is a pleasure to let you know that your manuscript is now accepted for publication in Life Science Alliance. Congratulations on this interesting work.

DISTRIBUTION OF MATERIALS:

Again, congratulations on a very nice paper. I hope you found the review process to be constructive and are pleased with how the manuscript was handled editorially. We look forward to future exciting submissions from your lab.

Sincerely,

Shachi Bhatt, Ph.D.

Executive Editor

Life Science Alliance

<https://www.lsjournal.org/>

Interested in an editorial career? EMBO Solutions is hiring a Scientific Editor to join the international Life Science Alliance team. Find out more here -

https://www.embo.org/documents/jobs/Vacancy_Notice_Scientific_editor_LSA.pdf